# Pharmacokinetic Interaction between Atorvastatin and Omega-3 Fatty Acid in Healthy Volunteers

**DOI:** 10.3390/ph15080962

**Published:** 2022-08-03

**Authors:** Jae Hoon Kim, Jung Sunwoo, Ji Hye Song, Yu-Bin Seo, Won Tae Jung, Kyu-Yeol Nam, YeSeul Kim, Hye Jung Lee, JungHa Moon, Jin-Gyu Jung, Jang Hee Hong

**Affiliations:** 1Chungnam National University Hospital Clinical Trials Center, Daejeon 35015, Korea; wogns420@gmail.com (J.H.K.); swj4991@cnu.ac.kr (J.S.); sjhye80@gmail.com (J.H.S.); seoyub24@gmail.com (Y.-B.S.); 2Department of Medical Science, Chungnam National University College of Medicine, Daejeon 35015, Korea; 3Translational Immunology Institute, Chungnam National University, Daejeon 35015, Korea; 4Korea United Pharm., Inc., Seoul 06116, Korea; wtjung@kup.co.kr (W.T.J.); kynam@kup.co.kr (K.-Y.N.); kys262@kup.co.kr (Y.K.); 5Caleb Multilab., Inc., Seoul 06745, Korea; hjlee@calebml.co.kr (H.J.L.); jhamoon@calebml.co.kr (J.M.); 6Department of Family Medicine, Chungnam National University Hospital, Daejeon 35015, Korea; 7Department of Pharmacology, Chungnam National University College of Medicine, Daejeon 35015, Korea

**Keywords:** dyslipidemia, docosahexaenoic acid, eicosapentaenoic acid, atorvastatin, drug-drug interaction, pharmacokinetics

## Abstract

The interaction between statins and omega-3 fatty acids remains controversial. The aim of this phase 1 trial was to evaluate the pharmacokinetics of drug-drug interaction between atorvastatin and omega-3 fatty acids. Treatments were once-daily oral administrations of omega-3 (4 g), atorvastatin (40 mg), and both for 14 days, 7 days, and 14 days, respectively, with washout periods. The concentrations of atorvastatin, 2-OH-atorvastatin, docosahexaenoic acid (DHA), and eicosapentaenoic acid (EPA) were determined with LC-MS/MS. Parameters of DHA and EPA were analyzed after baseline correction. A total of 37 subjects completed the study without any major violations. The geometric mean ratios (GMRs) and 90% confidence intervals (CIs) of the co-administration of a single drug for the area under the concentration–time curve during the dosing interval at steady state of atorvastatin, 2-OH-atorvastatin, DHA, and EPA were 1.042 (0.971–1.118), 1.185 (1.113–1.262), 0.157 (0.091–0.271), and 0.557 (0.396–0.784), respectively. The GMRs (90% Cis) for the co-administration at steady state of atorvastatin, 2-OH-atorvastatin, DHA, and EPA were 1.150 (0.990–1.335), 1.301 (1.2707–1.1401), 0.320 (0.243–0.422), and 0.589 (0.487–0.712), respectively. The 90% CIs for most primary endpoints were outside the range of typical bioequivalence, indicating a pharmacokinetic interaction between atorvastatin and omega-3.

## 1. Introduction

Atorvastatin lowers cholesterol levels in hepatocytes by inhibiting 3-hydroxy-3-methylglutaryl-coenzyme A reductase (HMG-CoA reductase), a rate-regulating step in cholesterol synthesis. It lowers low-density lipoprotein cholesterol (LDL-C) levels, increases high-density lipoprotein cholesterol (HDL-C) levels, and decreases triglyceride (TG) levels, thereby alleviating hypercholesterolemia and dyslipidemia [1]. Atorvastatin also reduces the incidence of non-lipogenic multifactorial atherosclerosis [2]. The clinical usefulness of statins has been shown through large-scale clinical trials such as the Scandinavian Simvastatin Survival Study (4S), Cholesterol and Recurrent Events (CARE), Long-term Intervention with Pravastatin in Ischemic Disease (LIPID), A Study to Evaluate the Effect of Rosuvastatin On Intravascular Ultrasound-Derived Coronary Atheroma Burden (ASTEROID), Justification for the Use of Statins in Primary Prevention: An Intervention Trial Evaluating Rosuvastatin (JUPITER), as well as a primary preventive effect for cardiovascular disease and a secondary preventive effect in patients who already have coronary artery disease [2,3,4,5,6]. Statins are effective but also have adverse effects; an example is rhabdomyolysis, an infrequent but fatal effect of statins. Other muscle problems include cramps, stiffness, tendonitis, and inflammatory myopathy. Non-muscle adverse effects include liver dysfunction and increased blood glucose levels [7].

Omega-3-acid ethyl esters 90 is marketed globally for secondary prevention and improving hypertriglyceridemia after myocardial infarction. Omega-3 reduces the amount of very low-density lipoprotein cholesterol (VLDL-C) by inhibiting TG production in the liver, thereby reducing the amount of TG in the blood [8]. In addition, it reduces thromboxane A2 production and prevents myocardial infarction through its antiarrhythmic and anti-inflammatory effects. A meta-analysis has also shown that omega-3 has no clinically significant adverse effects [9,10].

Lowering LDL-C levels is a primary objective for dyslipidemia management, and high-intensity statins (rosuvastatin, pitavastatin, and atorvastatin) are recommended as a treatment [11,12]. There is a residual risk of cardiovascular events when TG levels are uncontrolled despite LDL-C control. Thus, dual therapy for managing dyslipidemia accounted for 18.6% of total therapy in 2018, with a steady yearly increase in its use [13].

Statin + omega-3 is one of the most common dual therapies in Korea, together with statin + ezetimibe and statin + fibrate. Barter et al. [14] and Nambi et al. [15] reported that dual therapy with statins and omega-3 for CVD management in patients with mixed dyslipidemia could improve their lipid profiles without additional laboratory tests or increase the risk of adverse effects on the liver or muscle.

The Investigational Product (IP), UI018, is under development by the Korean United Pharm. Inc. (Seoul, Korea). A lower dose of UI018 was approved by the Korean Ministry of Food and Drug Safety (MFDS) in 2021 as a fixed-dose combination (FDC) of atorvastatin (10 mg) and omega-3-acid ethyl esters 90 (1000 mg). In the phase 3 study of UI018, non-high-density lipoprotein cholesterol (non-HDL-C), TG, and total cholesterol levels were significantly decreased, while the VLDL-C level was significantly increased in the UI018 administration group compared with those of the atorvastatin-only group. This lipid profile change was more potent in older persons and without diabetes. The safety profile was similar between the two groups [16].

The atorvastatin dose was set as 40 mg, which is the maximum recommended initial dose, and the omega-3 dose was 4 g, which is the maximum daily dose. The Food and Drug Administration recommends administering the highest dose that has proven safety in a DDI clinical trial [17].

The pharmacokinetic (PK) interaction of statins with omega-3 is still controversial, and so far, no study has revealed the PK interaction between atorvastatin and omega-3. Thus, the aim of this phase 1 study of UI018 was to report the pharmacokinetics of drug-drug interaction (DDI) between atorvastatin and omega-3 fatty acids. 

## 2. Results

### 2.1. Subject Demographics

Forty male subjects (Group A: *n* = 20; Group B: *n* = 20) were enrolled and randomized, and thirty-seven (Group A: *n* = 17; Group B: *n* = 18) completed the study without major violations. The mean age, height, and weight of the study subjects were 22.9/24.6 years, 175.9/175.3 cm, and 69.4/68.4 kg, respectively (Table 1).

### 2.2. Pharmacokinetics

The PK analysis set included a total of 37 subjects (Group A: *n* = 17; Group B: *n* = 18) who completed all study procedures without major violations. The mean plasma concentration–time profiles of atorvastatin, 2-OH-atorvastatin, docosahexaenoic acid (DHA), and eicosapentaenoic acid (EPA) at steady state are shown in Figure 1. The PK parameters are listed in Table 2. The geometric mean ratios (GMRs) and 90% confidence intervals (CIs) of the co-administration to the single drug administration in the area under the concentration-time curve during the dosing interval at steady state (AUC_τ,ss_) of atorvastatin, 2-OH-atorvastatin, DHA, and EPA were 1.042 (0.971–1.118), 1.185 (1.113–1.262), 0.157 (0.091–0.271), and 0.557 (0.396–0.784), respectively. The GMRs (90% CIs) of the co-administration to the single drug administration for maximum plasma concentration at steady state (C_max,ss_) of atorvastatin, 2-OH-atorvastatin, DHA, and EPA were 1.150 (0.990–1.335), 1.301 (1.207–1.401), 0.320 (0.243–0.422), and 0.589 (0.487–0.712), respectively. Except for the AUC_τ,ss_ of atorvastatin, 90% CIs for most of the primary endpoints were outside the range of typical bioequivalence (Table 3).

### 2.3. Safety

All 40 subjects (Group A: *n* = 20; Group B: *n* = 20) who were administered IP at least once were included in the safety analysis set. A total of twenty adverse events (AEs) were reported in 15 subjects. In the treatment group, eight AEs were reported in six subjects after administration of omega-3 alone, three AEs in two after administration of atorvastatin alone, and nine AEs in seven after co-administration of omega-3 and atorvastatin (Table 4). In addition, one serious adverse event (SAE) was reported after the administration of atorvastatin alone. The cause of the SAE was hospitalization to treat oral cellulitis due to molar caries. All AEs, including the SAE, were resolved without sequelae.

## 3. Discussion

This study showed PK DDI between atorvastatin (40 mg) and omega-3 (4 g). The 90% CIs for most primary endpoints were outside the range of typical bioequivalence, indicating a PK interaction between atorvastatin and omega-3. Safety outcomes were similar between atorvastatin or omega-3 single administration and atorvastatin + omega-3 co-administration. Until recently, no studies have revealed the PK profiles for the interaction between atorvastatin and omega-3.

In this study, the 90% CIs for most primary endpoints of atorvastatin and 2-OH-atorvastatin were outside the range of 0.8 to 1.25, indicating a PK interaction between atorvastatin and omega-3. However, Di Spirito et al. [18] showed that when omega-3 (4 g) and atorvastatin (80 mg) were co-administered to healthy adults, the AUC and C_max_ of atorvastatin did not change. This is consistent with the results of other studies that have also shown that the PK characteristics of statins are not changed by omega-3 [19,20].

Atorvastatin is metabolized by cytochrome P450 3A4 (CYP3A4); however, opinions on the effect of omega-3 on CYP3A4 differ. Some studies have shown that CYP3A4-catalyzed metabolism is inhibited by unsaturated fatty acids [21,22]. On the other hand, Hu et al. [23] reported that the increased levels of fatty acids upregulated CYP3A4 activity in a concentration-dependent manner. This discrepancy is considered owing to the high intersubject variability of atorvastatin concentrations, lower dosage, and the small sample size, other than the CYP3A4 mechanism.

The AUC_τ,ss_ of DHA and EPA decreased by 84.3% and 44.3%, respectively, after co-administration with atorvastatin (40 mg). The effects of statins on omega-3 levels vary between studies. Ciucanu et al. [20] showed that rosuvastatin decreased total fatty acid and free fatty acid levels but did not affect arachidonic acid (AA), EPA, and DHA levels. On the other hand, Nozue and Michishita showed that rosuvastatin and pitavastatin decreased the level of DHA and DHA/AA ratio but did not significantly change the EPA/AA ratio [24]. Nakamura et al. [25] and Harris et al. [26] showed that pravastatin and simvastatin significantly decreased the EPA/AA ratio. In addition, Kurisu et al. [27] showed that high-intensity statins decreased the levels of EPA and DHA. Although the mechanism of the effect of statins on omega-3 was not fully explained until recently, this study outcome is consistent with the results of studies that stated that statins might reduce omega-3 efficacy [28].

There were no clinically significant differences between single drug administration and atorvastatin + omega-3 co-administration in safety parameters, such as AEs, vital signs, physical examination results, clinical laboratory test results, and 12-lead ECGs. Atorvastatin (40 mg) and omega-3 (4 g) were well tolerated without serious myopathy in healthy male adults.

The change in the efficacy of atorvastatin after co-administration with omega-3 is not expected to be clinically significant because the GMR for AUC_τ,ss_ of 2-OH-atorvastatin was approximately 1. In addition, as DHA and EPA are endogenous substances, this reduction in concentration has little clinical significance unless the subject has an omega-3 deficiency.

## 4. Materials and Methods

### 4.1. Subjects

This clinical trial recruited healthy male volunteers aged between 19 and 45 years, weighing 55 kg or more, and with ideal body weight (IBW) within 20% of the value calculated as follows: IBW (kg) = (height [cm] − 100) × 0.9. All subjects were screened based on medical history, vital signs, physical examination results, clinical laboratory test results, and 12-lead electrocardiograms (ECGs). The exclusion criteria were HDL-C level < 35 mg/dL and a history or family history of myopathy. All volunteers provided written informed consent before the study procedures. The consent form was reviewed and approved by the MFDS and Institutional Review Board (IRB) of Chungnam National University Hospital (IRB NO. CNUH 2016-02-008-021).

### 4.2. IPs

Omacor Soft Cap (omega-3 acid ethyl ester 90 1 g, DHA 380 mg, EPA 460 mg, Kuhnil Pharmaceutical Co., Ltd., Seoul, Korea) and Lipitor Tab (atorvastatin 40 mg, Pfizer Korea Ltd., Seoul, Korea) were used as the products for the study.

### 4.3. Study Design

The clinical trial protocol was prepared according to the guidelines of the Korean Good Clinical Practices and the International Council for Harmonisation Guidelines. The clinical trial protocol was approved by the MFDS and IRB of Chungnam National University Hospital, and the study was registered at ClinicalTrials.gov (NCT05190133). This study was conducted in accordance with the principles of the Declaration of Helsinki. This is a randomized, open-label, multiple-dosing, two-arm, two-period, one-sequence study design implemented at the Chungnam National University Hospital Clinical Trials Center in Daejeon, Korea.

The subjects were randomized to either Group A or Group B at a 1:1 ratio. The scheme of the study design is presented in Figure 2. Treatments were conducted according to the group (Group A: Treatment A–Treatment C, Group B: Treatment B–Treatment C). Treatment A consisted of oral administration of omega-3 4 g once daily for 14 days. Treatment B was orally administered atorvastatin 40 mg once daily for 7 days. Furthermore, Treatment C consisted of oral administration of omega-3 4 g and atorvastatin 40 mg once daily for 14 days. All IPs were orally administered within 30 min of breakfast intake. In addition, to minimize bias, the intake of omega-3-rich foods, such as chub mackerel, tuna, and walnut, was restricted from seven days prior to blood sampling for PK analysis.

The time to reach a steady state was determined to be 14 days for Treatment A and 7 days for Treatment B, based on previous studies [29,30,31,32]. Similarly, the washout periods were determined to be 21 days for Treatment A and 14 days for Treatment B, which are more than five times the half-life of both drugs, based on previous studies.

Blood sampling was performed as follows: Group A: 24, 12, and 0 h prior to the first dose in each period for baseline measurement; day 12 (and day 47), 0 h; day 13 (and day 48), 0 h; day 14 (and day 49), 0 h (pre-dose); and 1, 2, 3, 4, 5, 6, 7, 8, 10, 12, and 24 h after the last multiple doses. Group B: day 5 (and day 33), 0 h; day 6 (and day 34), 0 h; day 7 (and day 35), 0 h (pre-dose); and 0.5, 1, 1.5, 2, 2.5, 3, 4, 6, 8, 10, 12, and 24 h after the last multiple doses. Blood samples were collected from EDTA K2 tubes. The collected blood samples were centrifuged at 1950–1970× *g* for 10 min at 4 °C. The plasma in the Eppendorf tubes was stored below −70 °C until analysis.

### 4.4. Determination of Plasma Concentrations

#### 4.4.1. Atorvastatin and 2-OH-Atorvastatin

The plasma concentrations of atorvastatin and 2-OH-atorvastatin were determined using a validated high-performance liquid chromatography-tandem mass spectrometer (HPLC-MS/MS) (HPLC: Agilent 1200 series system, Agilent, Germany; MS/MS: 6460 triple quadrupole MS/MS system, Agilent) [33]. An analytical column, Kromasil 100-3 5C18 (2.1 mm × 50 mm, AKZONOBEL, Sweden), was used as a stationary phase, and the mobile phase consisted of 5 mM ammonium formate (0.1% formic acid) and acetonitrile at a ratio of 40:60 (*v*/*v*) with a flow rate of 0.25 mL/min. The frozen sample was thawed at room temperature (approximately 21–24 °C), 200 µL was transferred to a borosilicate glass, and 50 µL of the internal standard (atorvastatin-d5, 50 ng/mL in methanol) was added. A total of 100 µL of 0.1 M ammonium acetate (pH 4.2) and 2 mL methyl tert-butyl ether (MTBE) were added, and the mixture was shaken for 5 min. The mixed samples were centrifuged at 4000 rpm for 5 min, and the supernatant was removed and evaporated to dryness under a nitrogen stream at 40 °C. The residue was dissolved in 100 µL of the mobile phase, and 4 µL was injected into the LC-MS/MS system. The concentrations of atorvastatin and 2-OH-atorvastatin were determined using positive electrospray ionization (ESI) and multiple reaction monitoring (MRM) modes. Ion transitions were *m/z* 559.3 to 440.2 and *m/z* 427.255 to 295.2 for atorvastatin and 2-OH-atorvastatin, respectively. The calibration curves were established linearly between 0.5 and 100 ng/mL, while R^2^ was over 0.99. The coefficient of variation (CV%) of the accuracy and precision of the plasma concentration measurements were all within 20%.

#### 4.4.2. DHA and EPA

Plasma concentrations of DHA and EPA were determined using a validated HPLC-MS/MS (HPLC: Acquity UPLC systems, Waters, USA; MS/MS: Xevo TQ-S, Waters) [34,35]. After transferring 25 µL of the samples to a microtube, 50 µL of the internal standard and 1000 µL of acetonitrile were added, vortexed for 10 s, and centrifuged at 14,000 rpm for 3 min. After centrifugation, 50 µL of 5 M hydrochloric acid was added to 200 µL of the supernatant and incubated at 80 °C for 30 min. After cooling, 200 µL was transferred to a test tube, 500 µL of distilled water and 1000 µL of MTBE were added, mixed by shaking for 5 min, and centrifuged at 4000 rpm for 5 min. After centrifugation, approximately 850 µL of the supernatant was transferred to a new tube and evaporated under a 50 °C nitrogen stream. The residue was dissolved in 300 µL of methanol, and 2 µL was injected into the LC-MS/MS system. An analytical column, Kinetex^®^ 1.7 µM C18 100 Å (50 mm × 2.1 mm), was used as the stationary phase, and the mobile phase consisted of 2 mM ammonium acetate in distilled water and acetonitrile at a ratio of 40:60 (*v/v*) with a flow rate of 0.3 mL/min. The concentrations of DHA and EPA were determined using the negative ESI and MRM modes. Ion transitions were *m/z* 327.05 to 283.15 and *m/z* 301.05 to 257.15 for DHA and EPA, respectively. The calibration curves were established linearly between 20 and 4000 µg/mL for DHA and 10 and 2000 µg/mL for EPA, and the R2 for both curves was greater than 0.99. The CV% of the accuracy and precision of the plasma concentration measurements were all within 20%.

### 4.5. Pharmacokinetic Analysis

The per protocol (PP) dataset was used to analyze the PK parameters. The PP dataset excluded subjects who had major protocol violations. The primary PK endpoints were AUC_τ,ss_ and C_max,ss_ for atorvastatin, 2-OH-atorvastatin, DHA, and EPA. The parameters of DHA and EPA levels were analyzed after baseline correction. Secondary endpoints included the minimum plasma concentration at steady state (C_min,ss_) and time to C_max,ss_ (T_max,ss_). PK analysis was performed by a non-compartmental method using WinNonlin^®^ software (version 6.3; Pharsight, Mountain View, CA, USA). AUC_τ,ss_ was calculated using a linear trapezoidal method. The C_max,ss_, C_min,ss_, and T_max,ss_ were directly determined from the observed concentration-time data. The baseline was calculated as the average concentration at 24, 12, and 0 h prior to the first dose in each period. Negative values after baseline correction were treated as zero for analysis.

### 4.6. Statistical Analysis

Statistical analyses were performed using the SAS^®^ software (version 9.4; SAS Institute, Cary, NC, USA). All PK parameters were summarized using descriptive statistics. Primary PK endpoints were presented as GMRs and 90% CIs of the co-administration of omega-3 and atorvastatin to the administration of omega-3 or atorvastatin alone. GMRs and 90% CIs were calculated using a linear mixed-effect model. The 90% CIs of the primary PK endpoints were evaluated against conventional bioequivalence criteria from 0.8 to 1.25.

### 4.7. Safety and Tolerability

Safety and tolerability were assessed based on AEs, vital signs, physical examinations, clinical laboratory tests (complete blood count; blood chemistry including liver function test; triglyceride, LDL, and HDL levels; urinalysis), and 12-lead ECGs. All AEs were coded using system organ classes and preferred terms in the Medical Dictionary for Regulatory Authorities (MedDRA) version 18.1.

## 5. Conclusions

This study revealed PK DDI between atorvastatin 40 mg and omega-3 4 g. However, the safety was similar between atorvastatin or omega-3 single administration and atorvastatin + omega-3 co-administration. UI018 efficacy was established in patients with dyslipidemia in a phase 3 study. In conclusion, UI018, atorvastatin 40 mg FDC, and omega-3 4 g could be new therapeutic options for uncontrolled mixed dyslipidemia. Although statin and omega-3 co-administration is effective for hyperlipidemia patients with omega-3 deficiency is known [36], there are still many controversies about omega-3 and statin combination therapy; therefore, additional large-scale phase 3 studies or big-data studies are needed. When physicians and pharmacists prescribe omega-3 and statins together, they need to determine whether the patient clinically requires the combination of both drugs.

## Figures and Tables

**Figure 1 pharmaceuticals-15-00962-f001:**
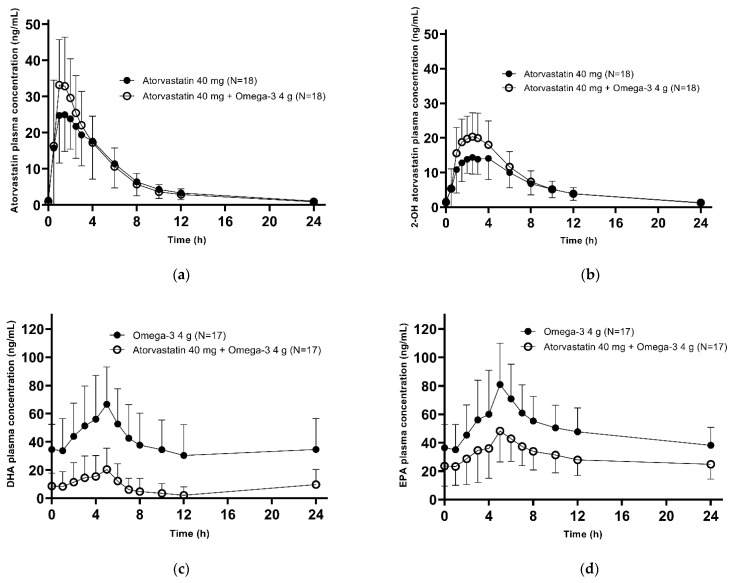
Mean plasma concentration-time profiles of (**a**) atorvastatin, (**b**) 2-OH-atorvastatin, (**c**) baseline-corrected DHA, and (**d**) baseline-corrected EPA after multiple oral administration as single drug administration vs. co-administration with a linear scale. The error bars represent the standard deviation.

**Figure 2 pharmaceuticals-15-00962-f002:**
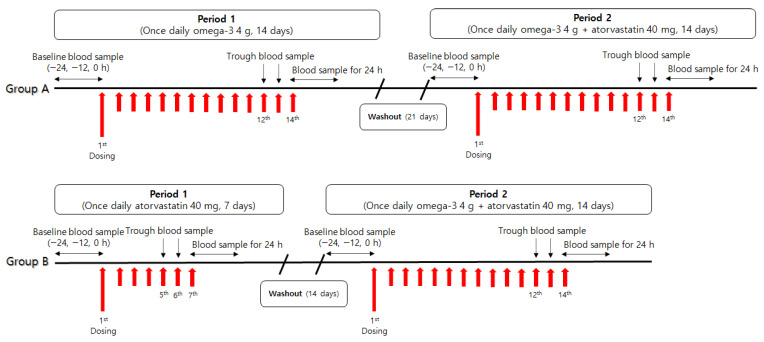
Study design.

**Table 1 pharmaceuticals-15-00962-t001:** Demographic characteristics of the subjects.

	Group A (*n* = 17)	Group B (*n* = 18)
Age (year)	22.9 ± 4.3	24.6 ± 3.5
Height (cm)	175.9 ± 4.7	175.3 ± 3.5
Weight (kg)	69.4 ± 7.5	68.4 ± 6.6

**Table 2 pharmaceuticals-15-00962-t002:** Pharmacokinetic parameters for atorvastatin, 2-OH-atorvastatin, DHA, and EPA after single drug administration or co-administration.

Parameter	Atorvastatin (*n* = 18)		2-OH-Atorvastatin (*n* = 18)		DHA (*n* = 17)		EPA (*n* = 17)	
	Alone	Combination	Alone	Combination	Alone	Combination	Alone	Combination
AUC_τ,ss_ (ng·h/mL)	170.6 ± 70.2	175.3 ± 65.0	139.1 ± 48.7	164.9 ± 56.1	910.5 ± 520.8	204.0 ± 211.0	1175.2 ± 399.6	721.3 ± 298.5
C_max,ss_ (ng/mL)	34.7 ± 10.7	40.0 ± 13.5	17.9 ± 6.0	23.1 ± 6.8	69.7 ± 27.6	22.4 ± 16.7	83.0 ± 27.9	51.4 ± 22.3
C_min,ss_ (ng/mL)	1.0 ± 0.7	0.7 ± 0.4	1.2 ± 0.6	1.1 ± 0.6	25.0 ± 20.9	2.4 ± 6.4	32.4 ± 15.8	21.1 ± 12.1
T_max,ss_ (h)	1.5 [0.5–4.0]	1.0 [0.5–2.0]	2.0 [0.5–4.0]	1.75 [0.5–3.0]	5.0 [3.0–7.0]	5.0 [3.0–6.0]	5.0 [3.0–7.0]	5.0 [0.0–7.0]

Notes: Data are shown as mean ± standard deviation, except for T_max,ss_, which is shown as median [minimum–maximum]. Mean data are presented as arithmetic means. Abbreviations: DHA, docosahexaenoic acid; EPA, eicosapentaenoic acid; AUC_τ,ss_, area under the concentration-time curve during a dosing interval at steady state; C_max,ss,_ maximum plasma concentration at steady state; C_min,ss_, minimum plasma concentration at steady state; T_max,ss_, time to maximum plasma concentration at steady state.

**Table 3 pharmaceuticals-15-00962-t003:** Comparison of the pharmacokinetic parameters of atorvastatin, 2-OH-atorvastatin, DHA, and EPA between single drug administration and co-administration.

Parameter	Reference	GMR	90% CI for GMR	
			Lower Limit (%)	Upper Limit (%)
Atorvastatin				
AUC_τ,ss_	Atorvastatin	104.2	97.1	111.8
Cmax,ss	Atorvastatin	115.0	99.0	133.5
2-OH-atorvastatin				
AUCτ,ss	Atorvastatin	118.5	111.3	126.2
Cmax,ss	Atorvastatin	130.1	120.7	140.1
DHA				
AUCτ,ss	Omega-3	15.7	9.1	27.1
Cmax,ss	Omega-3	32.0	24.3	42.2
EPA				
AUCτ,ss	Omega-3	55.7	39.6	78.4
Cmax,ss	Omega-3	58.9	48.7	71.2

Test: Omega-3 + Atorvastatin. Abbreviations: DHA, docosahexaenoic acid; EPA, eicosapentaenoic acid; AUC_τ,ss_, area under the concentration-time curve during a dosing interval at steady state; C_max,ss,_ maximum plasma concentration at steady state; C_min,ss_, minimum plasma concentration at steady state; T_max,ss_, time to maximum plasma concentration at steady state.

**Table 4 pharmaceuticals-15-00962-t004:** Summary of the adverse events related to treatment.

	**Omega-3 4 g**	**Omega-3 4 g + Atorvastatin 40 mg**
Group A	20 (100.0)	19 (100.0)
Epigastric discomfort	3 (15.0)	
Alanine aminotransferase level increased		1 (5.3)
Blood creatine phosphokinase level increased	1 (5.0)	2 (10.5)
Levels of transaminases increased		1 (5.3)
Dysuria	1 (5.0)	
Nasopharyngitis	2 (10.0)	2 (10.5)
Scratch	1 (5.0)	
	**Atorvastatin 40 mg**	**Omega-3 4 g + Atorvastatin 40 mg**
Group B	20 (100.0)	18 (100.0)
Hordeolum	3 (15.0)	
Diarrhea		1 (5.3)
Epigastric discomfort	1 (5.0)	2 (10.5)
Nasopharyngitis		1 (5.3)
Cellulitis	1 (5.0)	

Notes: All data are presented as the number of subjects (%).

## Data Availability

The datasets for this study will not be shared because it is possessed by Korea United Pharm., Inc., Seoul, Korea.

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
