# Peer review of "Pharmacokinetic Interaction between Atorvastatin and Omega-3 Fatty Acid in Healthy Volunteers"

_pharmaceuticals, 2022, doi:10.3390/ph15080962_

Round 1

Reviewer 1 Report

This study is a good extension of the research work group on the efficacy and application combination of atorvastatin 40 mg/ω-3 fatty acids 4 g. however, several concerns need to be addressed to fit for publication as follows:

1.      Title: remove 40 mg and 4 g.

2.      It is not preferable to begin sentences with abbreviations like that in line 56 UI018 …..etc.

3.      The writing style should be formal from the third-person perspective. Do not use we (E.g. lines 64 and 170).

4.      Table 3: delete the column of Test and add “Omega-3 + atorvastatin” to the table legend.

5.      Material and methods:

-          The description of experimental groups (Lines 188-195) is very confused and unclear. It is highly recommended to be rewritten and describe clearly the treatment of each group. Also, adding a diagrammatic figure will be of high value. The same for blood sampling (Lines 202-209).

-          The IPs section should be first before the heading “Study design”.

-          Add the reference for the methods of determination of plasma concentrations for all compounds.

-          In the safety and tolerability section: the authors mentioned "clinical laboratory tests". What are the tests performed?

Reviewer 2 Report

Kim et al. established a study ‘’Pharmacokinetic drug-drug interaction study between atorvastatin 40 mg and omega-3 fatty acid 4 g in healthy volunteers" which is a very useful investigation as many patients consume supplements such as omega 3 and statin drug family. It is a very interesting and useful study. Well-conducted and well written. I have a few suggestions to make your paper clearer for readers:

  1. In the Abstract section, background subsection, kindly add briefly the possible omega-3 - statins interactions.
  1. Also explain why you have chosen these doses: 3g+40mg.
  1. The authors mentioned that "The concentrations of atorvastatin, 2-OH-atorvastatin, docosahexaenoic acid (DHA), and eicosapentaenoic acid (EPA) were determined" shortly write the names of the techniques used for this purpose.
  1. At the end of the abstract section, add your recommendations for pharmacists and physicians.
  1. I suggest removing the first paragraph of the introduction and starting with a paragraph talking about the statin family and Omega 3 f.a.
  1. Write in the Intro a paragraph about the adverse effects of the statin family and Omega 3 f.a.
  1. Also, write about the pharmacokinetic effects of both and possible interactions.
  1. The conclusion part needs improvement, and you should write it to contain your study's major outcomes, your own conclusion, and your future recommendations for scientists for further investigations, physicians, and pharmacists to expand your paper's readers.
  1. To the reference list, and kindly updated references 2020-2022.

  1. The manuscript needs major grammar, typo and editing corrections

Round 2

Reviewer 1 Report

No further comments to be addressed

Reviewer 2 Report

The authors did all the required corrections, thank you